# Toward Information Theoretic Active Inverse Reinforcement Learning

**Ondrej Bajgar**
University of Oxford

**Dewi Sid William Gould**
Alan Turing Institute

**Jonathon Liu**
Independent

**Oliver Newcombe**
University of Oxford

**Rohan Narayan Langford Mitta**
Independent

**Jack Golden**
University of Oxford

## Abstract

As AI systems become increasingly autonomous, aligning their decision-making to human preferences is essential. In domains like autonomous driving or robotics, it is impossible to write down the reward function representing these preferences by hand. Inverse reinforcement learning (IRL) offers a promising approach to infer the unknown reward from demonstrations. However, obtaining human demonstrations can be costly. Active IRL addresses this challenge by strategically selecting the most informative scenarios for human demonstration, reducing the amount of required human effort. Where most prior work allowed querying the human for an action at one state at a time, we motivate and analyse scenarios where we collect longer trajectories. We provide an information-theoretic acquisition function, propose an efficient approximation scheme, and illustrate its performance through a set of gridworld experiments as groundwork for future work expanding to more general settings.

## 1 Introduction

Stuart Russell suggested three principles for the development of beneficial artificial intelligence: its only objective is realizing human preferences, it is initially uncertain about these preferences, and its ultimate source of information about them is human behavior [1]. *Apprenticeship learning* via Bayesian *inverse reinforcement learning* (IRL) can be understood as a possible operationalization of these principles: Bayesian IRL starts with a prior distribution over reward functions representing initial uncertainty about human preferences. It then combines this prior with *demonstration* data from a human expert acting approximately optimally with respect to the unknown reward, to produce a posterior distribution over rewards. In apprenticeship learning, this posterior over rewards is then used to produce a policy that should perform well with respect to the unknown reward function.

However, getting human demonstrations requires scarce human time. Also, many risky situations where we would wish AI systems to behave especially reliably may be rare in these demonstration data. Bayesian active learning can help with both by giving queries to a human demonstrator that are likely to bring the most information about the reward. Most prior methods for active IRL [2, 3, 4] queried the expert for action annotations of particular isolated states. However, in domains such as autonomous driving with a high frequency of actions, it can be much more natural for the human to provide whole trajectories – say, to drive for a while in a simulator – than to annotate a large collection of unrelated snapshots. There is one previous paper on *active IRL with full trajectories* [5] suggesting a heuristic acquisition function whose shortcomings can, however, completely prevent learning. We instead suggest using the principled tools of Bayesian active learning for the task.

Workshop on Bayesian Decision-making and Uncertainty, 38th Conference on Neural Information Processing Systems (NeurIPS 2024).

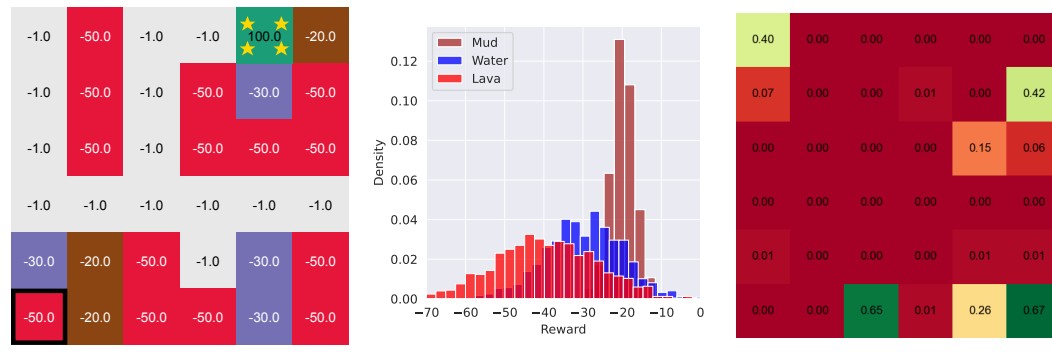

(a) Ground-truth rewards.   (b) Current belief over rewards.   (c) EIG of each initial state.

Figure 1: (a) shows an illustrative gridworld and its true rewards. The lower left corner has a "jail" state with negative reward from which an agent cannot leave. The starred green state is the terminal "goal" state with a large positive reward. The brown, blue, and red states are "mud", "water", and "lava" type states respectively, whose rewards are unknown to the IRL agent. The IRL agent tries to learn the rewards of these three state types from expert demonstrations. (b) shows the learned distributions over the rewards of the "mud", "water", and "lava" state types respectively, at some particular step of the active learning process. These learned reward distributions are used to calculate the EIG of obtaining another expert demonstration starting from each given state, shown in (c). In this case, a demonstration starting in the bottom right state gives the most information about the unknown reward parameters.

The article provides the following contributions: we formulate the problem of active IRL with full expert trajectories and adapt the expected information gain (EIG) acquisition function to this setting. We then provide an algorithm approximating the EIG and present experiments showing its superior performance relative to random sampling and two other baselines in gridworlds. We consider this initial investigation in tabular settings a stepping stone toward algorithms for more general settings.

## 2  Task formulation

Let $\mathcal{M}(\xi) = (\mathcal{S}, \mathcal{A}, p_\xi, r, \gamma, t_{\max}, \rho_\xi)$ be a parameterized Markov decision process (MDP), where $\mathcal{S}$ and $\mathcal{A}$ are finite state and action spaces respectively, $p_\xi : \mathcal{S} \times \mathcal{A} \to \mathcal{P}(\mathcal{S})$ is the transition function where $\mathcal{P}(\mathcal{S})$ is a set of probability measures over $\mathcal{S}$, $r : \mathcal{S} \times \mathcal{A} \to \mathbb{R}$ is an (expected) reward function,[1] $\gamma \in (0, 1)$ is a discount rate, $t_{\max} \in \mathbb{N} \cup \{\infty\}$ is the time horizon, and $\rho_\xi$ is the initial state distribution. The parameter $\xi$ will be used to set up the environment in active learning. Due to space limitations here, we present experiments where $\xi = s_0$ deterministically chooses an initial state, but our method can be used also for choosing the transition dynamics.

We assume we are initially uncertain about the reward $r$, and our initial knowledge is captured by a prior distribution $p(r)$ over rewards, which is a distribution over $\mathbb{R}^{|\mathcal{S}||\mathcal{A}|}$ – a space of vectors representing the reward associated with each state-action pair. We also have access to an expert that, given an instance $\mathcal{M}(\xi_i)$ of the MDP, can produce a trajectory $\tau_i = \left((s_0^i, a_0^i), \ldots, (s_{n_i}^i, a_{n_i}^i)\right)$, where $s_0^i \sim \rho_{\xi_i}$, $s_{t+1} \sim p_{\xi_i}(\cdot | s_t, a_t)$, and

$$\pi_E^\xi(a_t | s_t) = \frac{\exp(\beta Q_\xi^*(s_t, a_t))}{\sum_{a' \in \mathcal{A}} \exp(\beta Q_\xi^*(s_t, a'))} \ , \tag{1}$$

which is called a *Boltzmann-rational* policy, given the optimal Q function $Q_{\xi_i}^*$ and a hyperparameter $\beta$ expressing how close to optimal the expert behaviour is (where $\beta = 0$ corresponds to fully random behaviour and $\beta \to +\infty$ would yield the optimal policy).

---

[1]Our formulation permits for the reward to be stochastic. However, our expert model (1) depends on the rewards only via the optimal Q-function, which in turn depends only on the expected reward. Thus, the demonstrations can only ever give us information about the expectation. Throughout the paper, the learnt reward function can be interpreted either as modeling a deterministic reward, or an expectation of a stochastic reward.

The task of *Bayesian active inverse reinforcement learning* is to sequentially query the expert to provide demonstrations in environments $\xi_1, \ldots, \xi_N$ to gain maximum information about the unknown reward. We start with a (possibly empty) set of expert trajectories $\mathcal{D}_0$ and then, at each step of active learning, we choose a parameter $\xi_i$ for the MDP, from which we get the corresponding expert trajectory $\tau_i$. We then update our demonstration dataset to $\mathcal{D}_i = \mathcal{D}_{i-1} \cup \tau_i$, and the distribution over rewards to $p(r|\mathcal{D}_i)$, which we again use to select the most informative environment setup $\xi_{i+1}$ in the next step. We repeat until we exhaust our limited demonstration budget $N$.

Our goal can be operationalized as minimizing the entropy of the posterior distribution over rewards, once all expert demonstrations have been observed. This is equivalent to maximizing the log likelihood of the true parameter value in expectation, or to maximizing the mutual information between the demonstrations and the reward. We call this the *information-theoretic objective*.

For the *apprenticeship-learning objective*, we use the final posterior $p(r|\mathcal{D}_N)$ to produce an *apprentice policy* $\pi^A := \text{argmax}_\pi \mathbb{E}_r[\mathbb{E}_\tau[\sum_{s_t, a_t \in \tau} \gamma^t r(s_t, a_t)]]$ maximizing the expected return, where $\tau$ is a trajectory on a known target setup $\xi_\text{target}$ with $s_0 \sim \rho_{\xi_\text{target}}$, $s_{t+1} \sim p_{\xi_\text{target}}(\cdot|s_t, a_t)$ and $a_t = \pi(s_t)$.

# 3 Method

Our goal at each step is to select an environment setup $\xi$ that will produce the most information in expectation. In *Bayesian experimental design* (BED) [6], especially Bayesian optimization [7], this is often framed in terms of an *acquisition function* that for each $\xi$ estimates how useful it would be to select, i.e. we would like to select $\xi$ that maximizes the acquisition function.

We use the acquisition function most common in BED, the *expected information gain* (EIG):

$$EIG_n(\xi) = \mathbb{E}_{r|\mathcal{D}_n}\left[\mathbb{E}_{\tau|r,\xi}[\log p(r|\tau, \xi) - \log p(r)]\right] = \mathbb{E}_{r|\mathcal{D}_n}\left[\mathbb{E}_{\tau|r,\xi}[\log p(\tau|r, \xi) - \log p(\tau|\xi)]\right],$$

where the expectation over trajectories is taken with respect to $\rho_\xi$, $p_\xi$, and an expert policy that would correspond to the reward $r$ from the outer expectation, taken with respect to the current posterior.

In general, the expectations cannot be calculated analytically. A basic way to approximate the EIG would be using the following nested Monte Carlo estimator for each candidate environment setup $\xi$:

1. Sample $N_r$ reward functions $r_i$ from the current posterior $p(r|\mathcal{D}_n)$. For each $r_i$:
    (a) Sample $N_\tau$ trajectories $\tau_{ij}$ from the estimated expert policy $\hat{\pi}_E^{r_i, \xi}$ given the environment parameters $\xi$, where $\hat{\pi}_E^{r_i, \xi}$ would be the Boltzmann-rational policy corresponding to $r_i$.
    (b) Estimate[2] $p(\tau_{ij}|r_i, \xi) = \prod_{s_t, a_t \in \tau} \hat{\pi}_E^{r_i, \xi}(a_t|s_t)$ and $p(\tau_{ij}|\xi) = \frac{1}{N_r} \sum_k p(\tau_{ij}|r_k, \xi)$.
2. Approximate EIG using the Monte Carlo estimate:

$$\widehat{EIG}(\xi) = \frac{1}{N_r} \sum_{i=1}^{N_r} \frac{1}{N_\tau} \sum_{j=1}^{N_\tau} \left[\log p(\tau_{ij}|r_i, \xi) - \log p(\tau_{ij}|\xi)\right]. \tag{2}$$

While conceptually simple, the computational demands of this grow quickly with the size of the state space. Thus, in the next section, we discuss a method based on Bayesian optimization to allocate any computational budget we may have more efficiently.

## 3.1 Efficient sampling with Bayesian optimization

We propose to use *Bayesian optimization*, in particular the *upper confidence bound* (UCB) algorithm, to adaptively choose from which initial states to sample additional hypothetical trajectories to efficiently estimate the EIG. We still use the basic structure of (2), but instead of using the same number of samples in each initial state, we dynamically choose where to add additional samples to best improve our chance of identifying the state maximizing the EIG.

We model the information gain from each hypothetical trajectory $\tau_{si}$ starting in state $s$ as a Gaussian noisy observation of the true EIG value:

$$e_{si}(s) \sim \mathcal{N}(\mu_s, \epsilon_s^2), \tag{3}$$

---

[2]Note that we can omit the probabilities due to the initial state and transitions since these cancel out in Eq. 2.

where we assume $\mu_s = \text{EIG}(s)$. We also assume we have a prior on the mean and noise,

$$\mu_s \sim \mathcal{N}(\mu_{\text{prior}}, \sigma^2_{\text{prior}}), \quad \epsilon_s \sim p_\phi(\epsilon_s). \tag{4}$$

We first collect a fixed initial number of samples for each state. Then, we repeat the following until we have exhausted a budget of trajectories $T$. Following standard Gaussian updating, after an observation of a new hypothetical trajectory from $s$, we update the parameters

$$\mu_s = \left( \frac{\mu_{\text{prior}}}{\sigma^2_{\text{prior}}} + \frac{n_s \widehat{\text{EIG}}(s)}{\epsilon^2_s} \right) \left( \frac{1}{\sigma^2_{\text{prior}}} + \frac{n_s}{\epsilon^2_s} \right)^{-1}, \quad \sigma^2_s = \left( \frac{1}{\sigma^2_{\text{prior}}} + \frac{n_s}{\epsilon^2_s} \right)^{-1}, \tag{5}$$

where $n_s$ is the number of observed trajectories from $s$, and $\widehat{EIG}(s) = \frac{1}{n_s} \sum_{i=1}^{n_s} e_{si}$ is the average of the corresponding EIG estimates. We then update $\epsilon_s$ using maximum a posteriori estimation:

$$\epsilon_s = \arg\max_{\epsilon_s} \left[ p_\phi(\epsilon_s) \cdot \mathcal{N} \left( \widehat{\text{EIG}}(s) \mid \mu_s(\epsilon_s), \sigma_s(\epsilon_s) \right) \right]. \tag{6}$$

and compute a new EIG estimate for the value $s^*$ maximizing the upper confidence bound:

$$s^* = \arg\max_s \text{UCB}(s) := \arg\max_s \mu_s + \kappa \sigma_s, \tag{7}$$

where $\kappa$ is a UCB hyperparameter (we use $\kappa = 3$).

## 4 Experiments

We evaluated our EIG-based methods with full trajectories on two randomized gridworld setups against several simpler baselines: (1) uniform random sampling, (2) selecting the state with maximum entropy in Q-values, (3) querying just a single state (to measure the benefits of whole trajectories), and (4) selecting the starting state leading to trajectories with maximum posterior predictive entropy over the optimal policy. The last one is an acquisition function from [5], which is the only previous work on active IRL over trajectories that we are aware of.

We use two main metrics: the entropy of the posterior distribution over reward parameters after a given number of steps of active learning and the expected return (with respect to the initial state distribution and environment dynamics) of an apprentice policy maximizing this expected return (also with respect to the posterior over rewards).

We test on two kinds of gridworld environments: one with fewer state types (and thus reward parameters) than states, which gives the algorithm a known environment structure to exploit, and one with a single random reward per state. Full details on our experiments and additional results (including the efficiency gains from Bayesian optimization) are provided in Appendix C.

**Structured gridworld**   We begin with the $6 \times 6$ gridworld shown in Figure 1a. This environment is deterministic with 5 actions corresponding to moving in the four directions and staying in place. The agent can move freely, except for the bottom-left "jail" state, which is non-terminal, has a negative reward, and traps the agent permanently upon entry. In terms of the state rewards, there are five different state types and both the apprentice and the expert know the type of each state a priori. The rewards associated with two state types are known: "path" type, with a reward of $-1$, and a "goal" type with reward 100, which is also terminal. There are 3 state types, which we refer to as "water", "mud", and "lava", which have unknown negative reward. We place an independent uniform prior in the interval $[-100, 0]$ on the reward of each state type. Our goal is to infer the reward of these three state types.

**Fully random gridworld**   We also performed experiments on a $7 \times 7$ gridworld with each state's reward drawn from $\mathcal{N}(0, 3)$. Each state furthermore has a 10% probability of being terminal. States with reward above the 0.9 quantile of rewards are also terminal.

**Results**   Figure 2 shows results for structured environment (the results for the fully random environment can be found in Appendix C), comparing active methods with randomly choosing trajectories. We observe that the performance of EIG in terms of posterior entropy and in terms of apprentice performance is superior to the baselines. The Q-Entropy does better than random initially, but then starts to do worse due to repeatedly sampling from states with irreducible uncertainty. Notably, on

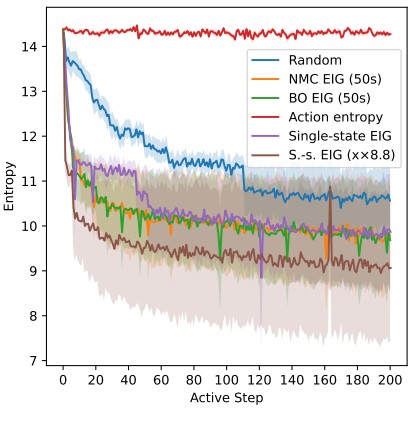

(a) Posterior entropy

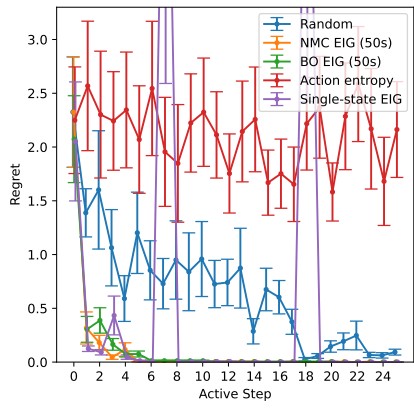

(b) Apprentice regret

Figure 2: Entropy of the posterior the regret of the apprentice policy on the structured environment. NMC stands for the naive nested Monte Carlo estimation, while BO stands for Bayesian optimization. "S.s. EIG (x×8.8)" denotes single-state EIG with the x axis scaled by 8.8 - the mean length of trajectories collected by the full trajectory EIG variants.

the structured environment, the posterior predictive action entropy acquisition function from [5] breaks entirely, as it only ever queries for demonstration trajectories that start in the jail state, as this state trivially has a uniform action distribution, and demonstrations starting in the jail state deterministically remain in the jail state. Thus these demonstrations offer no useful information about the expert reward or the policy. Based on these results, we believe that our information-theoretically derived acquisition function is more principled and robust.

## 5   Discussion and conclusion

We have provided a preliminary study of the problem of active IRL with full trajectories in tabular environments. We have shown that an information theoretic acquisition function provides improvements both in terms of achieving lower posterior entropy, and in terms of apprentice performance. It thus allows using the scarce time of demonstrators more efficiently. We see this preliminary study with synthetic gridworlds and demonstrations as a stepping stone toward an extension to continuous state spaces and more realistic settings.

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

## A   Related work

Our work builds on two strands of work: inverse reinforcement learning (IRL) and active learning, which we will address in turn. The IRL problem itself was introduced by Russell [8], preceded by the closely related problem of *inverse optimal control* formulated by Kalman [9]. See Arora and Doshi [10] and Adams et al. [11] for recent reviews of the already extensive literature on IRL. Ramachandran and Amir [12] introduced the Bayesian formulation of the problem that we build on here.

Active learning has first been introduced into IRL by Lopes et al. [2] who collect action annotations in states where the current posterior over reward functions implies high ambiguity about the expert's action. A key limitation of this approach is that the action ambiguity can come from several actions being equally good according to the true reward (such as in the jail state in our structured environment). This ambiguity remains even when we already have plenty of expert data for a given state, which can result in queries that do not bring any additional value. Our approach focuses on states that actually bring extra information about the uknown reward. A second limitation – common with other methods discussed below – is that the expert provides single action annotations. This is not practical in settings such as autonomous driving where actions are sampled with high frequency, and it may be more natural for a human demonstrator to provide longer trajectories (i.e. drive for a while) rather than give annotations for unrelated individual time frames.

Bueuning et al. [13] query full trajectories in the context of IRL, where the active component arises in a choice of transition function from a set of transition functions at each step of learning. Buening et al [14] also query full trajectories in a different context involving two cooperating autonomous agents.

Our work addresses the setting of identifying an optimal strategy for choosing trajectories within a fixed environment.

Instead of directly providing demonstrations, in Sadigh et al. [15], the human expert is asked to provide a relative preference between two sample trajectories synthesized by the algorithm. While this generally provides less information per query than our formulation, it is a useful alternative for situations where providing high-quality demonstrations is difficult for humans.

Brown et al. [3] present a risk-aware approach, which queries individual states with the highest $\alpha$-quantile policy loss, i.e. the states with a high risk that the apprentice action could be much worse than the expert's action.

Instead of querying at arbitrary states, Losey and O'Malley [16] and Lindner et al. [17] synthesize a policy that explores the environment to produce a trajectory which subsequently gets annotated by the expert. We instead let the expert produce the trajectory.

The closest baseline for our work, and the only existing work we are aware of that deals with full trajectories in active IRL, comes from Kweon et al. [5]. Like in our experimental setup, they propose an acquisition function for querying for expert trajectories starting from a given initial state. Their acquisition function is based on maximizing the posterior predictive action entropy along the

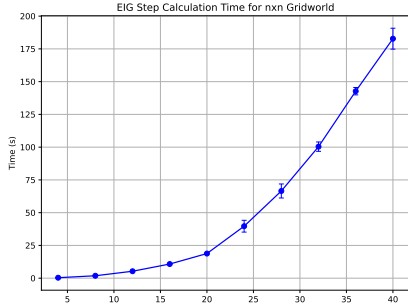
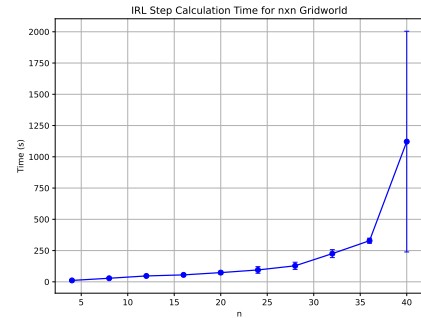

(a) Scaling of EIG step calculation in time (s) with increasing grid size.

(b) Scaling of IRL step calculation in time (s) with increasing grid size.

Figure 3: These plots show that the EIG calculation step scales approximately quadratically in $n$, or linearly in the number of steps, and is very consistent. For comparison, the plot also shows the scaling of the time required to run the PolicyWalk algorithm for Bayesian IRL

demonstration trajectory $\tau$. That is, maximizing

$$
\mathbb{E}_{\tau \sim \hat{\pi}_E^D} \left[ \sum_{t=0}^{|\tau|} \mathcal{A}(s_t) \middle| s_0 \right] \tag{8}
$$

where

$$
\mathcal{A}(s) = \sum_a -\hat{\pi}_E^D(a|s) \log \hat{\pi}_E^D(a|s),
$$

i.e. the entropy of the estimated expert policy $\hat{\pi}_E^D$ at state $s$, estimated from demonstration data $D$.

## B  Scaling Properties

We also provide a brief view of the scaling properties of the nested Monte Carlo estimation of the EIG with respect to increasing sizes of a scaled-up version of the structured environment. We ran 3 repeated trials with adaptive step sizes, 50 warmup steps and 200 samples, for 5 active learning steps, then timed the computational time for the EIG calculation as well as the associated PolicyWalk algorithm for Bayesian IRL. The results are displayed in Figure 3. They suggest that the Bayesian IRL algorithm may be the limiting factor in scaling up, though the ValueWalk algorithm (which we found not suitable for the structured environment, but performing well on the fully random one) generally displays better scaling properties [18]. That said, for scaling the algorithms further, especially to continuous spaces, we expect to need to resort to methods based on variational inference.

On the other hand, we observed the Bayesian-optimization-based method of EIG calculation to scale more favourably, since it does not need to assign a uniform budget to all $n^2$ squares, but can focus only on the most promising ones based on an initial estimate. In our initial investigation, we found that assigning a quarter of the budget across initial squares (which still scales quadratically) and scaling the rest linearly tended to preserve performance on par with nested Monte Carlo with a fully quadratic scaling of the budget, but we are leaving a fuller analysis to a future version of this paper.

## C  Experiment details

### C.1  Bayesian IRL methods

Our active learning uses a Bayesian IRL method as a key component. In our experiments, we used two methods based on Markov chain Monte Carlo (MCMC) sampling: on the structured environment, we used PolicyWalk [12], while on the environment with a different random reward in every state, we used the faster ValueWalk [18], which performs the sampling primarily in the space of Q-functions

before converting into rewards. We also tried a method based on variational inference [19], but we found its uncertainty estimates unreliable for the purposes of active learning.

For MCMC sampling, we used Hamiltonian Monte Carlo [20] with the no-U-turns (NUTS) sampler [21] and automatic step size selection during warm-up (starting with a step size of 0.1). At every step of active learning, we ran the MCMC sampling from scratch using all demonstrations available up to that point. We ran for 100 warm-up steps and then 200 collected samples. For subsequent usage, we generally thin the sample to 50 samples to reduce autocorrelation.

## C.2 Baselines

We compare our methods against various baseline approaches. To evaluate the value of using *full trajectories* in our EIG estimate, we also give results for experiments where $\widehat{\text{EIG}}$ is computed after querying a single state only (equivalently a unit-length trajectory), and the returned demonstration has unit length. Relatedly, we consider a baseline experiment in which $N = 8$ single states are queried at each active step, where $N$ equals the average length of demonstrations in the active setting. In a sense, the latter baseline serves as an "upper bound" of performance we could hope to achieve with a fixed budget of trajectory lengths. For these length-one trajectories, we otherwise use the same EIG calculation as for our longer trajectories.

For the Q-entropy baseline, we calculate the optimal Q-value corresponding to each reward sample (this is in fact produced as a byproduct of both Bayesian IRL algorithms) and estimate its entropy using the $k$-nearest-neighbours method with $k = 5$. We then select the state with maximum entropy as the next initial state. The reasoning behind this is that uncertainty in the expert policy is directly dependent on the uncertainty in the Q-value. Furthermore, the uncertainty in the Q-value captures not only uncertainty about the reward in the given state, but also about the rewards and Q-values of states that are expected to follow.

For the posterior predictive action entropy baseline [5], we use the acquisition function (8), while adapting everything else to be consistent with our experiments. Specifically, this means calculating the estimated expert policy $\hat{\pi}_E^{\mathcal{D}}$ directly from samples of the expert Q-values and assuming a Boltzmann rational expert. The expectation in Eq. 8 is again approximated by sampling a number of trajectories starting from $s_0$, according to policy $\hat{\pi}_E^{\mathcal{D}}$. For the structured environment, this acquisition function only queries trivial trajectories remaining on the jail state which did not terminate, so it was necessary to truncate these trajectories a maximum length. We chose 15 for this maximum.

## C.3 Experimental setup

In sampling the hypothetical trajectories, we cap their length at 15 in the structured environment and 10 in the fully random one. In approximations, we used 20 reward samples and 2 sampled trajectories per reward.

### C.3.1 Structured environment

In the structured environment, we use an expert rationality coefficient of $\beta = 1$. We do not provide any initial demonstrations. All experiments were run with 10 random reward assignments, consistent across all tested methods. The reward was drawn from the same prior as was used by the Bayesian IRL method, i.e. independent Uniform$[0, 100]$ for the 3 rewards associated with the 3 state types.

### C.3.2 Fully random environment

In the fully random environment, we use an expert rationality coefficient of $\beta = 1$ and provide 1 initial trajectory starting in the top left corner. Each method was run with 16 random reward and terminal-state assignments. The reward was drawn from the same prior as was used by the Bayesian IRL method, i.e. $\mathcal{N}(0, 3)$, i.i.d. for each state.

## C.4 Additional results

In Figure 4 we provide the results on the fully random environment. To make the entropy plot more legible, we aggregate the results for 10 steps subsequent steps, i.e. the result at 10 steps is the mean

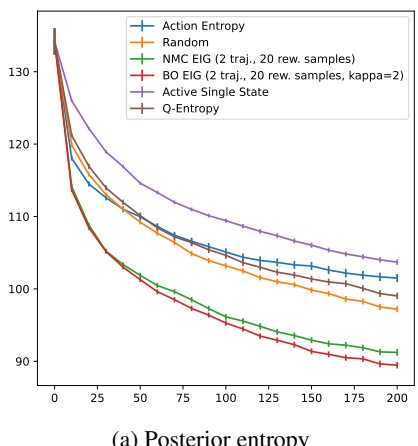

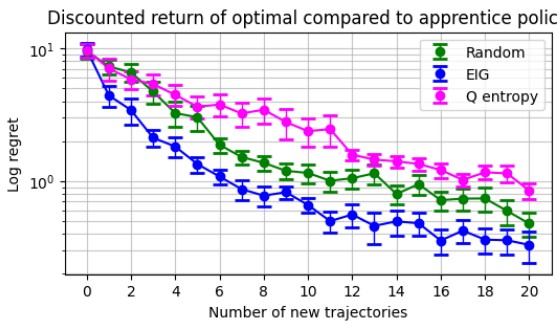

(a) Posterior entropy

(b) Apprentice policy cumulative regret, on the 7x7 random, averaged over 16 experiments.

Figure 4: Results on the fully random environment. NMC stands for the naive nested Monte Carlo estimation, while BO stands for Bayesian optimization.

and standard error across steps 1-10, at 20 across 11-20 and so on. The result at 0 is the performance based on only the initial trajectory (before the first active step).

The results are mostly consistent with the results on the structured environment with several differences. Here we do not observe any advantage to sampling using the Q-entropy baseline even in the early steps. We also observe an advantage of the Bayesian-optimization calculation for EIG - with the same budget, the method is able to achieve better posterior entropy – in fact, a similar entropy is achieved by the naive nested Monte Carlo estimation only with about double the budget. With the same budget of trajectory samples, the Bayesian optimization method incurs a less than 10% increase in computation time, taking 6.6 instead of 6.2 seconds per step on average. On top of this, the Bayesian IRL method takes about 20 seconds to collect the 200 reward samples (+ 100 warm up steps).

