# OpenReview forum: "Toward Information Theoretic Active Inverse Reinforcement Learning"
_NeurIPS.cc/2024/Workshop/BDU — NeurIPS BDU Workshop 2024 Poster_

### Official Review · Reviewer_6ksw · 2024-09-27
**Interesting problem, method not explained sufficiently**

**Rating:** 4
**Confidence:** 3

**Review:**

## Strengths

The motivation for this work was very nice, and the abstract and introduction
had me excited to read further. I agree with the idea that once you query an
expert, it is often acceptable/practical to query the expert for more than one
action at a time, which is largely overlooked in the literature, and could
reduce the amount of times an expert is &ldquo;invoked&rdquo; in practice.


## Weaknesses

The main weakness of the paper, to me, is the lack of clarity in the technical
sections. I made an honest effort to understand how their method works,
especially given that I found the problem really well motivated (see
strengths). Unfortunately, despite my efforts, I can only at best guess how
their method works. Weaknesses are listed below.

1.  Section 3.1 is extremely unclear. In particular, the way it&rsquo;s written, it is
    not at all clear that $\widehat{EIG}(s)$ is computed by equation (2) (given
    that the notation is different, and there is no reference to reinforce this).
    It should be made *much* more clear how the tools from Section 3.1 are
    integrated with the framework discussed at the beginning of Section 3.
    My best guess is that you use Section 3.1 to pick the next source state $s$ to
    query an expert trajectory from. Then, perform steps 1 and 2 (line 74) with
    trajectories starting from $s$ to estimate $\widehat{EIG}(s)$. Then you use
    this new estimate to update the posterior on state-wise EIGs, which informs
    the next source state to query with. Ultimately, this UCB strategy lets you
    essentially not repeat steps 1, 2 for each initial state (to find the optimal
    $\xi$) before each expert query, but steps 1, 2 are still necessary. When
    reading the paper, I thought Section 3.1 would somehow make steps 1, 2 more
    efficient, which caused confusion.
2.  The steps 1 and 2 also are very unclear. Step 1a says to sample
    trajectories from the &ldquo;estimated expert policy $\hat{\pi}^{r\_i, \xi}\_E$&rdquo; &#x2013; how
    do you estimate the expert policy? This is an important step that appears to
    be neglected. Clearly this cannot actually be an estimation of the
    expert, since it depends on a sampled reward function $r\_i$. I&rsquo;m assuming
    $\hat{\pi}^{r\_i, \xi}\_E$ is the optimal Boltzmann-rational policy for $r\_i$
    (maybe can be found by DP?). But
    then where are the expert demonstrations influencing anything?
    Again, my best guess is that $\tau_{ij}$ is actually sampled from the real
    expert, but that&rsquo;s not at all what&rsquo;s written, so I don&rsquo;t know.
3.  I can&rsquo;t tell if such a method can actually hope to be scaled. I&rsquo;m not
    necessarily criticizing the paper for not demonstrating results on larger
    MDPs, but I think the paper can benefit from a discussion about some
    potential avenues for scaling the method up to larger problems, since it is
    not at all obvious that this can be done.

The paper is over the page limit (slightly).


## Minor Issues and Questions

I found the exposition of the parameterized MDP confusing / unexpected; on first
read I didn&rsquo;t understand its purpose altogether. I think it could be very
helpful to give some examples of $\xi$ beyond just the indication of an initial
state. My understanding currently is that one could query the expert, for
example, at MDPs that force the expert into certain regions of the state space
that would otherwise be difficult to reach, and this is done through $p_\xi$. This
is fine since you are only trying to recover the reward function and not the
value function.

Why isn&rsquo;t the apprentice policy also a Boltzmann-rational policy?

On line 88, what does it mean to collect samples for each state? What are the
samples? Is this referring to expert trajectories starting at each state?

In Figure 2b, I don&rsquo;t think this is plotting regret (which would be
non-decreasing).

In Figure 1, it would be helpful to know which states in Figure 1a correspond to
the different cell types &#x2013; currently it is difficult to interpret how Figures
1a, 1b, 1c relate to one another without this information.

---

### Official Review · Reviewer_dDgv · 2024-10-04
**Leveraging full trajectories in inverse reinforcement learning**

**Rating:** 8
**Confidence:** 3

**Review:**

This paper addresses inverse reinforcement learning (IRL), in which the reward function (aligning with human behavior) is initially unknown/unspecified; Bayesian IRL starts with a prior distribution over reward functions which is then combined with data (coming from a demonstration by a human expert) to produce a posterior distribution over rewards. The key contribution of this paper is to use entire trajectories of actions from a human expert rather than one step at a time; for instance, in self-driving, a human expert could drive for some time instead of indicating just the next action to take. The authors formulate the problem as Bayesian active learning with full trajectories, including an adaptation of the EIG, and present two estimators (based on nested Monte Carlo and Bayesian optimization). In a structured tabular/gridworld environment, the authors show gains from using their methods in terms of posterior entropy and apprentice regret compared to Q Entropy and random sampling (further results are included in the appendix). This appears to be an original and potentially significant work in IRL.

The paper is for the most part clearly written and appears to contribute to the area of IRL. However, it would have been nice to see results comparing full trajectories to single state (these are only shown in the appendix), as it seems this would underscore one of the main contributions of the paper. It is also not clear why one would prefer BO EIG over NMC EIG as they appear to perform very similarly in the experiments, and this is not discussed by the authors. I also suggest better explanation of the figures; in Figure 1, the caption includes “mud”, “water”, “lava” (which is never referenced anywhere else in the paper), and in Figure 2, the caption uses acronyms which were never defined (BO EIG and NMC EIG; of course, I can guess what these mean, but initially I had to go back through the paper to try to look for the definition).

---

### Official Review · Reviewer_Q9XX · 2024-10-08
**A novel active IRL method using full trajectories via approximated EIG. While the approach is interesting, the experimental evaluation is limited and unclear.**

**Rating:** 6
**Confidence:** 3

**Review:**

**Summary**: This paper proposes an approach for active inverse reinforcement learning (IRL) from full trajectories instead of single-state queries. Expected information gain (EIG) is used as the acquisition function. EIG is estimated using upper confidence bound (UCB) algorithm, which renders the method efficient. Method is evaluated in two grid-world experiments. It is presented that the proposed approach over performs selected baselines in terms of entropy and regret minimization.

**Strengths:**
1. The idea of using full trajectories instead of single-state queries is interesting, and using approximated EIG for this purpose is novel to my knowledge.
The proposed acquisition function is explained well overall.

**Weaknesses:**
1. Experiments are very limited to basic settings and are difficult to understand. First, it is necessary to provide a demonstrative image of the environment in Figure 1 that clearly shows which tiles are lava/mud/water tiles. Second, what is 'the current belief over rewards'? Is it the initial belief or something else? It would also be meaningful to present sampled expert trajectories at different steps to show how this approach works differently from the baselines.
2. Figure captions should be much more informative. For instance, what does NMC EIG mean in Figure 2?
3. There is another method in the literature [1] that also claims to propose an active IRL approach that works with trajectories. A comparison between the proposed approach and [1] should be conducted. Also, as a baseline, instead of random sampling, [1] or a more advanced query-based active IRL method could be used.

[1]: Kweon, Sehee, Himchan Hwang, and Frank C. Park. "Trajectory-Based Active Inverse Reinforcement Learning for Learning from Demonstration." 2023 23rd International Conference on Control, Automation and Systems (ICCAS). IEEE, 2023.

---

### Decision · Program_Chairs · 2024-10-09

**Decision:**

Accept (Poster)

**Comment:**

This is a well-motivated work with a novel formulation of doing IRL with a full trajectory. For the camera ready, please take into account the reviewer's suggestions, especially on clarity.